REGISTERED REPORT PROTOCOL

# The price penalty for red meat substitutes in popular dishes and the diversity in substitution

Dominic Lemken *

Department of Agricultural Economics and Rural Development, University of Göttingen, Göttingen, Germany

* dlemken@gwdg.de

## Abstract

Life cycle assessments (LCA) often highlight the environmental and health benefits for consumers if western diets substitute red meat. However, the specific trade-off consumer face when asked to substitute a red meat dish is scarcely researched, often neglecting the bouquet of substitution options and/or the price component involved. Four substitution strategies are evaluated within an individually adapted choice based conjoint: the substitution by (1) the same red meat dishes with a halved meat portion size, (2) novel plant-based products that mimic the functionality and taste, (3) authentic plant-based components that just mimic the functionality, and (4) vegetarian dishes that just neglect the meat component if still familiar to consumers. The analysis is executed for three popular red meat dishes to account for consistency across meal scenarios, namely Meatballs, Spaghetti Bolognese and Sausage Buns. The analysis is sensitive to red meat consumption habits to better understand the preferences of consumers that can actually substitute a red meat intake.

## Introduction

I enjoy eating beef, but I fear we have to eat less. Beef production requires roughly 36 (±13) times the feed mass than the edible meat generated [1]. The conversion rate points to the amount of land resources required to provide food for human consumption. The carbon footprint of 1 calorie derived from beef is estimated at 22 g CO2-equivalent in contrast to 0,05 g for 1 calorie derived from pulses [2]. The widely noted EatLancet report considered the health and environmental impacts of diets and recommends roughly 2.6 kg of beef or lamb per year [3], while within the EU-27 about 15 kg are available for consumption each year [4]. From the report follows that the current level of red meat consumption is not sustainable, threatens our health and depletes our means of production. Nevertheless, red meat consumption is deeply rooted in western dietary patterns [5]. Consumers are often faced with tasteful red meat variations inherent to western food cultures.

This status of red meat in western food cultures makes red meat prevalent, especially in food settings that allow only for a limited number of offerings such as when eating-out. About 7% of all EU-expenditures are spend on the out-of-home food market [6]. The share is growing each year [6]. This out-of-home market is characterized by suppliers that have to prioritize

**Data Availability Statement:** All relevant data from this study will be made available upon study

completion. Once the project has been completed, the data will be made available via the GRO.data repositorium (https://data.goettingen-research-online.de). GRO.data is managed by the University of Göttingen and publicly accessible.

**Funding:** D.L. - The article is fully financed by the DFG project „Key food choices and climate change", project no. 431972934 (German Research Association, https://www.dfg.de/en/index.jsp). The DFG had no role in study design, data collection and analysis, decision to publish, or preparation of the manuscript.

**Competing interests:** no competing interests to declare.

food options, as opposed to a supermarket that approximately offers 20000 products. Such suppliers rightfully select popular dishes that the consumer is willing to pay a reasonable amount for. In consideration of red meat's externalities, the question needs to be addressed how much consumers are less willing to pay for substitutes and consequentially how much worse off are out-of-home suppliers if they start to prioritize substitutes.

Climate researchers and the life-cycle assessment (LCA) literature have analysed foods with respect to its environmental impact and the utility of a change, but with little consideration for adequate substitutes and often failed to understand consumer's trade-offs [7]. On an aggregated dietary level there is a rich literature on the drivers of adoption of purely vegetarian diets (literature reviews by [8,9]), including an increasing number of studies on strategies that may help to nudge consumers towards plant-based, i.e. vegetarian diets (literature review by [10]). However, the vast majority of consumers does not identify with a diet that strictly prohibits red meat. In line with dietary recommendations, such as the EatLancet report, consumers should be supported in cutting back on red meat intake not necessarily in stopping altogether. So, diets low in meat or daily choices against meat enable consumers' transition to healthier and more sustainable food choices, optionally vegetarian diets at some point. In this context, we analyse consumer behaviour on a dish or ingredient level.

A few studies have analysed meat preferences within dish compositions. These studies evaluated specific substitution options that we group in four categories:

Reduced meat portion size (a): Globally, reduced meat portion sizes are more popular than western food culture might suggest. In East Asia, the servings of meat are an essential part of the food culture while the quantity is low in most dishes, i.e. meat fulfills a flavouring function rather than providing calories to human consumption [11]. With respect to consumer acceptance, a meat reduction in Mexican cuisines adversely affects consumer liking, which does not hold for Indian cuisines [12]. This confirms the food culture component when designing substitutes. As international cuisines are increasingly adopted in western countries, the promotion of some of the cuisines presents an opportunity. However, it also highlights the difficulty of reducing portion sizes of existing western dishes. In a Dutch restaurant setting, a reduction from 210 to 180 g of meat in a meat-centred dish preserved most of the satisfaction previously experienced [13]. This points to the importance of subtle changes in order to let "meat reduced portion sizes" grow on consumers. Generally, caterers and restaurants are often reluctant to reduce the share of red meat, because they feel their customers judge the quality of the food based on the quantity of meat.

Novel plant based meat resembling products (b): The market for textured meat substitutes (2) in western Europe is growing at more than 10% annually since 2014 (Euromonitor 2018). Such substitutes can be integrated with ease in common meat dishes [14], as they hardly require recipe changes or any new preparation skills. Nevertheless, option (b) is a niche market unfamiliar to most consumers [15], often assumed to provide inferior taste [16]. A study on burgers estimated the average WTP for a beef burger around 13.5 USD, while the plant-based texture meat burger received an average WTP of 4.25 USD [17]. Although the product concepts of (b) are well accepted among meat eaters, considerable reformulation efforts will still be necessary to attract more consumers and convey an acceptable "meaty flavor" [18].

Authentic plant-based products (c): Consumers concept of authenticity is not always identical between food cultures. It includes rather a lower amount of processing and the plant based component has typically an established role within a dish composition. These components do not mimic the taste of meat but can obtain a similar functional role in the dish [19]. For example, Falafel may substitute Meatballs and grain burgers may replace meat-burger patties. Here, functionality refers to the way the dish is designed to be well balanced in the eyes of consumers. The functional component does not necessarily provide a nutritional profile that resembles

the meat component. Many variations of the option (c) exist for decades but have not been able to mainstream into the dietary patterns of meat-eaters in the west. Nevertheless, several consumer segments prefer the idea of meat substitution with authentic alternatives over meat resembling products [15].

Familiar vegetarian meal that simply neglects the meat (d): In some instances, the meat component is just neglected (4) and the dish remains familiar and authentic to consumers. Consumers are used to these dishes but they are often thought of as less appealing or low budget alternatives.

Conclusively, none of the substitution strategies is currently on an eye level with red meat consumption. The substitution categories have also been described in more detail in a mapping review by Boer and Aiking [5]. The review highlights how change strategies should build on the availability of a range of substitution options in order to give consumers flexibility. Unfortunately, consumer researchers have predominantly focused on just one category and denied consumers flexibility in the substitution task. A parallel analysis can highlight relative potentials, particularly with heavy meat eaters that are asked to substitute. Consumer research needs to better identify the type of consumers willing to substitute with any category, with a specific one and with none. Additionally, most substitution studies neglect the economic perspective, such as estimating WTP differences, which can make it difficult to communicate the results to policy makers and caterers.

This study fills the gap and accounts for a multitude of options to substitute. An utility framework for an individually adapted choice based conjoint (CBC) will be applied, where consumers are asked to decide between the set of substitution options and an applicable popular red meat dish. The analysis explores substitution preferences across three popular dishes, i.e. three case studies. Further, we calculate the carbon footprint of each evaluated dish to account for trade-offs between environmental benefits and consumers' WTP. The diversity of the substitution categories is typically not available in the out-of-home market that needs to prioritize dishes. A better understanding of the price penalty that substitutes invoke can help progressive food business and researchers to design more promising interventions on who and how to enable red meat substitution.

## Research objective

Based on the described substitution pathways two main research objectives are addressed: 1. An estimation of the price penalty (lower average WTP) for the substitution options, 2. An estimation of the price penalty sensitive to past consumption levels of the red meat dish at hand. While the former is of interest to caterers and restaurants that want to maximize revenues, the latter emphasizes frequent eaters that can actually substitute the red meat dish. Thereby, it provides a societal perspective on the potential of substitution, if substitutes were readily available in the market. Empirical evidence on the WTP for substitution pathways is still limited. The out-of-home market predominantly supplies the red meat dishes. A WTP analysis estimates beef burgers at almost three times the WTP of plant based burgers [17]. Other studies have shown how authentic alternatives or reduced meat portions can reduce meal satisfaction or sensory liking [12,13,18]. Therefore, we hypothesize a price penalty for all the substitution options (Table 1, H1), which is more pronounced for heavy red meat eaters because higher consumption of red meat is associated with a lower acceptance of substitutes [15] (Table 1, H3).

As we have previously explained how the preferences between substitution options are hardly understood, yet, we speculate that the price penalty is not equal for all substitution options (Table 1, H2). This implies an ordinal consumer preference structure, which also

**Table 1. Design planner.**

| Question | Hypothesis (H1 to H4) | Sampling plan | Analysis plan | Interpretation given different outcomes |
|---|---|---|---|---|
| Is there a price penalty for red meat substitution options? | U(RMD) > U(S1,S2,S3, S4) | Quoted consumer sample representative of age, gender and income | Model 1 | confirmed if consistently shown for all 3 RMD cases |
| Is the price penalty less severe for any of the substitution options? | U(S1,S2,S3,S4)≠ U(S1,S2, S3,S4) | | Model 1 | confirmed if at least one substitution option provides consistently greater or lesser utility across all 3 RMD cases |
| Is the price penalty more pronounced if consumers frequently eat the red meat dish? | μ[U(RMD) -U(S1,S2,S3, S4)]> U(RMD)- U(S1,S2, S3,S4) | | Model 1 and 2 | confirmed if consistently shown for all 3 RMD cases |
| Is the price penalty less severe for any of the substitution if consumers frequently eat the red meat dish? | μU(S1,S2,S3,S4)≠ μU(S1, S2,S3,S4) | | Model 2 | confirmed if at least one substitution option provides consistently greater or lesser utility across all 3 RMD cases |

U(X) = average utility derived from option X, S1-4 = substitution option 1 to 4, RMD = popular red meat dish, μ = a weighting matrix defined by the frequency consumers eat the RMD.

holds for the preference structure of heavy red meat eaters (Table 1, H4). An ordinal structure would be of interest to caterers and researchers faced with designing substitution interventions. However, it should be complemented by an idea of the type of consumer segments interested in a specific substitution category. We supplement the research objectives by exploratory research on the relationship of consumer characteristics with a preference for each substitution option. Each hypothesis is confirmed if a statistically significant difference (5%-level) is consistently found across all 3 cases of popular red meat dishes (Table 1).

Further, the carbon footprint of each dish is approximated, as one environmental indicator that urges the substitution of red meat. The consideration of the CO2-footprint helps to indicate whether a CO2 tax, which is broadly discussed, would be able to compensate for the price penalty between any of the dishes. Therefore, the estimated WTP results are complemented by a report of the CO2 tax-level demanded to level consumer preferences.

## Study design

The proposed study is a cross sectional survey experiment in the form of an individually adapted choice based conjoint, where participants evaluate hypothetical food choices in an out-of home lunch setting. Participants evaluate substitution options against popular red meat dishes, while the design is sensitive to heterogenous WTP among consumers.

### Three popular red meat dishes

To apply the previously presented substitution categories to tangible cases of food choices, we introduce three popular red meat dishes (RMD):

(a) Spaghetti Bolognese, as a "mixed dish" for which meat is not the main meal component [5]. The quantity of meat is not overly visible and consumers do not necessarily recognize a reduction of meat within the sauce. (b) Meatballs with rice and peas, as a "meat-centered" dish where plant-based ingredients are typically considered a side-dish [5], and (c) buns with butter and sausage, which presents a snack option for lunch instead of a warm meal decision (Table 2). The red meat dishes are well known. European consumers have developed a habitual use of them.

For the given dishes CO2 footprint estimations are available (Table 2). The IFEU Institute supplied CO2-footprint averages for the major ingredients of the dishes based on comparable value chains, capturing emission from farm to retail [20]. This allows for a calculation on

**Table 2. Experimental dishes and their CO2-footprint per portion.**

| Main ingredients/weight per portion (RDM) | CO2- footprint kg CO2 eq/kg | Red meat Dish | Substitute (1) | Substitute (2) | Substitute (3) | Substitute (4) |
|---|---|---|---|---|---|---|
|  |  | Spaghetti Bolognese | . . . ½ minced meat Bolognese | . . . Soy-based minced meat | Lentil Bolognese | Spaghetti Napoli |
| minced beef/166 g | 9,2 | 1,53 | 0,77 | X | X | X |
| tomato puree/100 g | 1,8 | 0,18 | 0,18 | 0,18 | 0,09 | 0,23 |
| pasta/166g | 0,7 | 0,12 | 0,12 | 0,12 | 0,12 | 0,12 |
| lentils/50 g | 1,2 | X | X | X | 0,06 | X |
| textured vegetable protein (soy)/166g | 1,0 | X | X | 0,17 | X | X |
| **Total** | **N.A.** | **1,83** | **1,06** | **0,46** | **0,27** | **0,34** |
|  |  | beef topping for 2 buns with butter | ½ thick-ness of topping . . . | vegan sausage topping . . . | Sliced Emmentaler cheese . . . | *N.A.* |
| bun/2x50 g | 0,7 | 0,07 | 0,07 | 0,07 | 0,07 | *0,07* |
| beef topping/2x20 g | 7,9 | 0,32 | 0,16 | X | X | *X* |
| butter/2x 20 g | 9 | 0,36 | 0,36 | 0,36 | 0,36 | *0,36* |
| vegan sausage/2x20g | 1,7 | X | X | 0,07 | X | *X* |
| Emmentaler cheese/2x20g | 6 | X | X | X | 0,24 | *X* |
| **Total** | **N.A.** | **0,75** | **0,59** | **0,50** | **0,67** | ***0,43*** |
|  |  | beef meatballs with rice and peas | ½ portion meatballs . . . | Soy-based patties . . . | Falafel . . . | *N.A.* |
| beef meatballs/200g | 9,2 | 1,84 | 0,92 | X | X | *X* |
| soy patty/200g | 1,1 | X | X | 0,22 | X | *X* |
| canned chickpeas/200g | 1,3 | X | X | X | 0,26 | *X* |
| Rice/50 g | 3,1 | 0,155 | 0,155 | 0,155 | 0,155 | *0,31* |
| canned Peas/50 g | 1,7 | 0,085 | 0,085 | 0,085 | 0,085 | *0,17* |
| **Total** | **N.A.** | **2,08** | **1,16** | **0,46** | **0,50** | ***0,48*** |

Portion sizes are informed by recipes from a popular German cooking website: Chefkoch.de, pictures are sourced from pixabay.com (only for non-commercial use), CO2 footprints are based on estimations of the IFEU-institute [20]. Minor ingredients and means of food preparation are neglected.

whether a CO2 tax-level can change the ordinal preference structure. Different CO2 tax-levels are globally discussed, e.g. 55 € per CO2 ton (5,5 cents per kg) is planned in Germany for the energy and transportation sector [21].

## Sampling procedure

A quoted consumer survey will be launched online. Participants shall be representative of Germans with internet access by age, gender and income [22]. Participants are invited to the study via a market research company who deals out a small reimbursement fee. The recruitment process allows to not inform about the particular focus of the study to avoid self-selection bias. Data quality checks include completeness and streamlining. Streamlining ensures that participants use a minimum of time on a set of questions [15]. Moreover, it monitors how participants progress through question blocks in order to exclude participants with speeding and/or systematic answering behavior. The monitoring of overall time spend on the survey becomes unnecessary. No other data exclusion criteria are planned. Budgetary constraints allow for 1000 consumers. Power calculation have been executed with a Cox model in STATA comparing a regression slope (the marginal utility for the substitution options) to a reference (the opt-out). Under the assumption of a minimum detectable effect size of 0.2, standard deviation of

0.5, power of 0.8 and a significance threshold of 0.05, 785 consumers should be surveyed. We aim for 800 valid responses. The survey will be coded with the Editix XML-Editor.

**Ethical approval.** The investigation is part of a work package in the research project "Key food choices and climate change". The full project proposal has been filed for ethical approval. The ethical committee of the University of Göttingen has granted the project ethical approval on the 18.12.2018.

## Choice procedure

After socio-demographic questions, consumers are introduced to the out-of-home setting. Consumers are asked to imagine a lunch routine where a facility was preselected and now they may select one out of five dishes. The dish components are named but the dishes are no further introduced because the unbiased consumer beliefs and opinions are of interest to the research objective. Before consumers face the choice sets, they are exposed to a cheap talk that explains how they should imagine the purchase situation, act as if real money would be asked of them and how consumers sometimes misinform on their preferences. Participants are required to confirm that they understood the cheap talk. The cheap talk reduces hypothetical bias [23].

The choice options are the substitution categories and the popular RMD (Table 3). Consumers are exposed to six choice tasks for each RMD (Table 3). In total 18 choice sets (6 choice-sets per choice scenario x 3 choice scenarios) are evaluated, which seems acceptable given that consumer fatigue should be lower because different meal scenarios are surveyed.

The choice-sets include only four attributes to reduce the burden on consumers. Many attributes can cause information overload so that consumers develop decision rules based on just one or two attributes, which results in poor estimations of the remaining attributes. The four attributes are: the dish (5 levels), waiting time (3 levels), CO2-footprint (3-levels) and price (4 levels within IACBC). The choice alternatives are labelled with the dish names. Each dish occurs once per choice set (alternative specific constant). Price levels are essential to the WTP analysis and waiting time represents the value of time which matters particularly in lunch settings and hints at a convenience factor of interest to all kinds of substitution interventions. The CO2-footprint indicates just how relevant the simple naming of a footprint advantage is to the task of reducing the price penalty. The popular red meat dish is treated as no-

**Table 3. a Attribute Design of choice based conjoint with 5 meal options (d-efficient design).** b Attribute Design of choice based conjoint with 4 meal options (d-efficient design).

| Choice set | 1 | 1 | 1 | 1 | 1 | 2 | 2 | 2 | 2 | 2 | 3 | 3 | 3 | 3 | 3 | 4 | 4 | 4 | 4 | 4 | 5 | 5 | 5 | 5 | 5 | 6 | 6 | 6 | 6 | 6 |
|---|---|---|---|---|---|---|---|---|---|---|---|---|---|---|---|---|---|---|---|---|---|---|---|---|---|---|---|---|---|---|
| Dish [ASC] | 1 | 2 | 3 | 4 | 5 | 1 | 2 | 3 | 4 | 5 | 1 | 2 | 3 | 4 | 5 | 1 | 2 | 3 | 4 | 5 | 1 | 2 | 3 | 4 | 5 | 1 | 2 | 3 | 4 | 5 |
| Wait time | 0 | 2 | 1 | 0 | 0 | 2 | 2 | 2 | 0 | 0 | 0 | 2 | 2 | 1 | 0 | 2 | 2 | 2 | 1 | 0 | 1 | 2 | 0 | 0 | 0 | 1 | 2 | 0 | 0 | 0 |
| CO2 footprint | 0 | 0 | 1 | 2 | 1 | 0 | 2 | 2 | 2 | 1 | 0 | 1 | 1 | 2 | 1 | 2 | 2 | 2 | 0 | 1 | 1 | 0 | 2 | 0 | 1 | 2 | 2 | 0 | 1 | 1 |
| f.Price | 3 | 0 | 0 | 1 | 3 | 1 | 3 | 3 | 3 | 3 | 3 | 0 | 0 | 0 | 3 | 2 | 2 | 2 | 0 | 3 | 0 | 0 | 2 | 0 | 3 | 3 | 0 | 3 | 0 | 3 |
| Choice set | 1 | 1 | 1 | 1 | 2 | 2 | 2 | 2 | 3 | 3 | 3 | 3 | 4 | 4 | 4 | 4 | 5 | 5 | 5 | 5 | 6 | 6 | 6 | 6 | | | | | | |
| Dish [ASC] | 1 | 2 | 3 | 4 | 1 | 2 | 3 | 4 | 1 | 2 | 3 | 4 | 1 | 2 | 3 | 4 | 1 | 2 | 3 | 4 | 1 | 2 | 3 | 4 | | | | | | |
| Wait time | 1 | 0 | 0 | 0 | 2 | 1 | 0 | 0 | 2 | 2 | 0 | 0 | 2 | 2 | 0 | 0 | 0 | 2 | 0 | 0 | 2 | 0 | 0 | 0 | | | | | | |
| CO2 footprint | 0 | 2 | 2 | 1 | 0 | 2 | 1 | 1 | 0 | 2 | 0 | 1 | 1 | 2 | 0 | 1 | 2 | 0 | 1 | 1 | 2 | 0 | 1 | 1 | | | | | | |
| f.Price | 0 | 3 | 0 | 3 | 1 | 3 | 0 | 3 | 2 | 2 | 2 | 3 | 0 | 2 | 1 | 3 | 3 | 0 | 0 | 3 | 0 | 3 | 0 | 3 | | | | | | |

d-efficiency = 1.206, waiting time [0 = 5 min, 1 = 10 min, 2 = 15 min], flexible price component in € [0 = 1, 1 = 2, 2 = 3, 3 = 4], dish [substitution option 1 to 4], co2 footprint [0 = high, 1 = not provided, 2 = low].

d-efficiency = 1.075, waiting time [0 = 5 min, 1 = 10 min, 2 = 15 min], flexible price component in € [0 = 1, 1 = 2, 2 = 3, 3 = 4], dish [substitution option 1 to 3], co2 footprint [0 = high, 1 = not provided, 2 = low].

**Table 4. Product attributes and attribute levels in the IACBC.**

| Attribute | Description | Levels |
|---|---|---|
| Dish | reduced meat portion size | ½ minced meat Bolognese, ½ thickness of topping, ½ portion of meatballs |
| | plant based meat resembling product | Soy based minced meat, Vegan sausage topping, Soy-based patties |
| | authentic plant-based product | Lentil Bolognese, Sliced Emmentaler cheese, Falafel |
| | just neglecting the meat component in | Spaghetti Napoli |
| | the RMD (reference) | (1) Beef sausage, (2) Spaghetti Bolognese, (3) Beef meatballs |
| Waiting time | Standard (reference) | 5 min |
| | Long | 10 min |
| | Very long | 15 min |
| CO2-footprint | Claim of a low CO2-footprint | Low |
| | No information provided | ? |
| | Claim of a high CO2-footprint | High |
| Price | The substitutes are subject to base and flexible price components set in IACBC design flexible price has 4 level for each dish, see also Table 6 | Total price for RMD: (1) 3 €, (2) 8 €, (3) 9 € Base price levels for all substitutes to: (1) 1.5 €, to (2) 4 €, to (3) 4 € |

purchase option with constant attributes, therefore the utility derived from substitutes is anchored to the utility derived from the red meat dish. Altogether, 15 levels of 4 product attributes are evaluated (Table 4).

The design of the attribute levels in each choice set is calculated with a STATA implementation of a d-efficient design based on a modified Fedorov-algorithm [24]. The STATA code is available upon request. The design assumes an utility penalty for each substitution dish in order to allow sensitive estimations at this range of values. Further, the design allows for the alternative specific constants and a fixed red meat dish option (Table 3). The choice-sets are followed by a set of questions on consumer characteristics.

## The choice pricing mechanism—Individually adapted choice-based conjoint (IACBC)

We apply an individually adapted choice-based conjoint (IACBC) [25]. Regular choice-based conjoint often suffer from extreme response behavior, i.e. a substantial number of consumers have always or never selected the no-purchase option. Consequently, there is no information on when these consumers are willing to accept a substitute or when they would stop buying the substitute, which results in poor WTP estimations of such consumers. For this study, there is little indication of how low a price level has to be to persuade frequent meat-eaters to endorse a substitute. IACBC accommodates heterogeneity with very low and very high consumer WTP. IACBC discounts the prices of all the substitution options by a specific factor whenever the no-purchase option is selected and multiplies prices with a factor greater than 1 whenever one of the substitution options is selected. Additionally, the prices for the substitutes oscillate.

To illustrate the IACBC price algorithm, imagine a scenario of a consumer willing to buy any substitute at a price of 6 € or lower. The popular red meat dish is always offered at 8 €, which is a 2 € price premium. Such a consumer will select a substitute if the price drops below 6,01 €. In theory, the IACBC is also equipped to detect WTP differences between just the substitutes but is simplified here. The price of the dishes consists of two components. The base

**Table 5. Illustration of IACBC approach for a consumer willing to pay up to 6 € for any substitute.**

| | | | | Final Price Option 1 to 4 | | | | |
|---|---|---|---|---|---|---|---|---|
| choice-set | base Price, option 1 to 4 | flexible price, option 1 to 4 | multiplier | option 1 | option 2 | option 3 | option 4 | choice |
| 1 | 4/5/5/4 | 2/3/4/5 | 1 | 6 | 8 | 9 | 9 | option 1 |
| 2 | 4/5/5/4 | 5/4/3/2 | 2 | 14 | 13 | 11 | 8 | 5 (RMD) |
| 3 | 4/5/5/4 | 5/2/3/4 | 1,2 | 10 | 7,4 | 8,6 | 8,8 | 5 (RMD) |
| 4 | 4/5/5/4 | 4/3/2/5 | 0,72 | 6,88 | 7,16 | 6,44 | 7,6 | 5 (RMD) |
| 5 | 4/5/5/4 | 4/5/2/3 | 0,432 | 5,73 | 7,16 | 5,86 | 5,3 | option 4 |
| 6 | 4/5/5/4 | 3/2/5/4 | 0,648 | 5,94 | 6,3 | 8,24 | 6,59 | option 1 |

RMD = popular red meat dish, flexible price assigned depending on CBC design (Table 3A and 3B).

price level presents a stable minimum price level for each substitute, which is always added to the flexible price. The flexible price component is multiplied with a multiplier that follows the function:

$$f(n)^z = (1 + 2/(n + 1))^z \qquad (1)$$

where n represents the number of shifts in direction between purchase and no-purchase decisions. z equals 1 if a substitute is selected and -1 if the no-purchase option is selected. The multiplier is set to 1 in the first choice-set. If a substitute is selected in choice-set 1, the multiplier is set to 2 and asymptotically converges to 1 (Table 5). For example, in choice-set 1 (Table 5) a substitution option has been selected, so that in choice-set 2 the price for option 1 equals: base price (4 €) + flexible price (5 €) x multiplier (2) (Table 5).

The IACBC is implemented with the methodological toolbox of dise-online (Dynamic Intelligent Survey Engine) [26]. The choice-design of the attribute levels creates unique sets and ensures regular price changes (flexible price component) between each set to capture preferences between substitution options (Table 4).

## Variables

The consumption of the RMD is controlled for with a frequency scale ranging from never to at least once a week, which has previously been applied to pulses that are not a daily meal component for any consumer [27]. The RMD consumption is needed to assign less weight to consumers currently not consuming the RMD which the upcoming section explains. Other explanatory variables complement the utility function to better understand the type of consumer with preferences for a specific form of substitution. The analysis can point out consumers more or less willing to accept a specific substitute. We intend to explore the relationship with the following food related characteristics: a. Beef-eating habits, b. Experience with the substitutes, c. Attitude towards out-of-home dining, d. Convenience factor in lunch settings, e. Attitude towards the red meat dishes, f. Price sensitivity and g. Socio-demographics.

The measurement of these concepts is summarized in Table 6 and resemble previous applications. For example, attitude towards out of home dining will be operationalized with the following items, "Going out for lunch is regularly part of my eating habits" and "I do not consider it luxury to go out to have lunch in a restaurant", which have been introduced within the food related lifestyle concepts by Grunert et al. [28]. The concept allows to control for general preferences with respect to out of home consumption, the scenario consumers are exposed to.

**Table 6. Consumer characteristics.**

| Concept | Base concept | Items | Scale |
|---|---|---|---|
| a. beef eating habits | The frequency scale is abstracted from [27] | The items ask for specific beef products to support participants memory of common beef product when evaluating their own consumption. The frequency for the following consumption items is addressed:<br>• Minced beef<br>• Beef Burger<br>• Beef steak<br>• Beef rips<br>• Beef ham<br>• Beef salami<br>• Any beef | 1 = never<br>2 = less than once a year<br>3 = a few times a year<br>4 = Once a month<br>5 = two or three times a month<br>6 = at least once a week |
| b. Experience with substitutes | Abstracted from Experience with product category [29] | Product category is a placeholder for each of the substitute dishes: I would have described myself as being very familiar with this product category:<br>• Lentil Bolognese<br>• Soy based minced meat Bolognese<br>• Spaghetti Napoli<br>• Vegan sausage<br>• Sliced Emmentaler cheese<br>• Soy based meat balls<br>• Falafel | 1 (definitely disagree) to 5 (definitely agree) |
| c. Attitude towards out-of-home dining | Abstracted from Food related lifestyle scale [28] | The following items are applied<br>• Going out for lunch is regularly part of my eating habits<br>• I do not consider it luxury to go out to have lunch in a restaurant | 1 (completely disagree) to 7 (completely agree), |
| d. Convenience factor in lunch settings | Abstracted from convenience orientation [30] | • It is important to me that I receive my lunch in a reasonable time span<br>• It is important to me that the lunch options are simple to select | 1 (definitely disagree) to 5 (definitely agree) |
| e. Attitude towards red meat dish | Abstracted from Attitude towards Food product [31] | All red meat dishes are evaluated concerning three attitudes towards the dish:<br>• Taste<br>• Socially appropriate<br>• expensive | Semantic differentials 5 point scale:<br>Tastes bad–tastes good<br>Not socially–socially<br>Inexpensive—Expensive |
| f. Price sensitivity | Abstracted from price sensitivity scale [32] | Respondents were asked to "think of the out-of-home purchase situation and rate their agreement with 3 items:<br>• I am willing to make an extra effort to find a low price for a meal<br>• I will change what I had planned to buy in oder to take advantage of a lower price for a meal<br>• I am sensitive to differences in price | 1(completely disagree) to 7 (completely agree) |

## Data analysis

The IACBC is consistent with random utility theory [33]. Consumer "n" derives utility from the choice alternative "a" in a given choice set "t". We apply a random parameter logit model, similar to Van Loo, Caputo, and Lusk [17] analysis of preferences for lab grown and plant based beef. The following utility function will be estimated:

$$U_{nat} = \mu_a + x_{nat}\beta_n + z_{nt}\delta_a + \varrho_{nat} \qquad (2)$$

where, $\mu_a$ is an alternative specific constant indicating utility derived from dish "a" relative to the red meat dish (baseline). $\beta_n$ are random parameters for the marginal utility derived from $x_{nat}$, a vector for price, waiting time, and the $CO_2$-footprint. This allows for consumer heterogeneity with respect to preferences, i.e. how consumers value a price penalty or a reduced waiting time. If the model cannot converge, waiting time, $CO_2$-footprint, and if needed price, can be estimated as fixed parameters. $\delta_a$ are fixed, alternative-specific coefficients on $z_{it}$, a vector of case-specific variables (income, gender, experience with substitutes, etc.) that describes consumer characteristics associated with preferences for each dish. $\varrho_{iat}$ is a random or unobservable term that follows a type I extreme value distribution. The probability that consumer "n"

chooses alternative "a" in choice set "t", conditional on the random parameters, can be written:

$$P_{nat}(\beta) = \frac{e^{\mu_a + x_{nat}\beta_n + z_{nt}\delta_a}}{\sum_{a=1}^{A} e^{\mu_a + x_{nat}\beta_n + z_{nt}\delta_a}} \qquad (3)$$

From the probability model, the utility for each dish will be calculated. Further, consumers are questioned on the frequency they consume the reference dish, i.e. the popular red meat dish. The response will be used for a subgroup analysis or rather to increase the weight of consumers that heavily consume the red meat dish. The weighting mechanism leads to a second model specification that better accounts for the preferences of consumers that can actually substitute. In a nutshell, the model (1) estimates the utility explained by substitution choices and consumer characteristics based on Eq 1. Model (2) repeats the estimation while weighting consumers by how often they eat the red meat dish. For both models, we will show the marginal effect sizes, p-values, not just the threshold level, and confidence intervals. A postestimation test for seemingly unrelated regressions can further indicate significant differences for coefficients between model specification 1 and 2.

In case of low discriminatory power between the utility derived from the substitutes, i.e. similar WTP for each substitution pathway, a latent class regression approach can be applied to understand the heterogeneity between consumer clusters.

## Limitations

When working with stated preferences, it should be acknowledged that food choices are not necessarily conscious decisions. Stated preferences suffer from contextual biases. However, a rational endorsement in a stated preference scenario can be expected to lead to less resistance towards the substitute during food intervention scenarios. Observed and stated preferences are known to correlate. The explored choice tasks do not account for substitution behaviour between different types of cuisines. Consumers may feel like Spaghetti Bolognese, but are swayed by the price of an unrelated substitute, e.g. a salad. The full complexity of food decision making cannot be simulated. The relationship between consumer characteristics and the WTP for substitution options will be pointed out but should not be interpreted as causal inference. A causal analysis of consumer characteristics is limited through the study properties of cross-sectional survey data. Prices, waiting time, and $CO_2$-footprint are manipulated between participants and choice set to allow for causal conclusions.

## Acknowledgments

I am thankful to Dorothee Verena Seybold for assisting in the design of the experiment and to Christian Schlereth for his help with the empirical implementation of this project.

## Author Contributions

**Conceptualization:** Dominic Lemken.

**Data curation:** Dominic Lemken.

**Formal analysis:** Dominic Lemken.

**Funding acquisition:** Dominic Lemken.

**Investigation:** Dominic Lemken.

**Methodology:** Dominic Lemken.

**Project administration:** Dominic Lemken.

**Resources:** Dominic Lemken.

**Software:** Dominic Lemken.

**Supervision:** Dominic Lemken.

**Validation:** Dominic Lemken.

**Visualization:** Dominic Lemken.

**Writing – original draft:** Dominic Lemken.

**Writing – review & editing:** Dominic Lemken.

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
