## [Decision Letter · Decision Letter 0]

25 Feb 2021

PONE-D-20-36544

Meatballs, Bolognese and Sausage-buns – the price penalty for red meat substitutes in popular dishes

PLOS ONE

Dear Dr. Lemken,

Thank you for submitting your manuscript to PLOS ONE. After careful consideration, we feel that it has merit but does not fully meet PLOS ONE’s publication criteria as it currently stands. Therefore, we invite you to submit a revised version of the manuscript that addresses the points raised during the review process.

We look forward to receiving your revised manuscript.

Kind regards,

Damian Adams

Academic Editor

PLOS ONE

Additional Editor Comments:

Thank you very much for submitting your paper to PLOS ONE. Two qualified experts have reviewed your paper, and were supportive of its focus, but identified several concerns that would need to be addressed before the paper could be published. In my opinion, both reviewers provide thoughtful suggestions, which should be fully addressed. Please be sure to carefully and clearly indicate how you have treated each of the comments in your response, and be sure that your revised manuscript fully reflects the needed edits. Thank you,

Damian Adams

Journal Requirements:

5. We note that one of the reviewers requests results and discussions sections. Please note that this request can be ignored, as it does not apply for this article type.

Reviewers' comments:

Reviewer's Responses to Questions

**Comments to the Author**

1. Does the manuscript provide a valid rationale for the proposed study, with clearly identified and justified research questions?

Reviewer #1: Partly

Reviewer #2: Partly

2. Is the protocol technically sound and planned in a manner that will lead to a meaningful outcome and allow testing the stated hypotheses?

Reviewer #1: Partly

Reviewer #2: Partly

3. Is the methodology feasible and described in sufficient detail to allow the work to be replicable?

Reviewer #1: Yes

Reviewer #2: Yes

4. Have the authors described where all data underlying the findings will be made available when the study is complete?

Reviewer #1: Yes

Reviewer #2: Yes

5. Is the manuscript presented in an intelligible fashion and written in standard English?

Reviewer #1: Yes

Reviewer #2: Yes

6. Review Comments to the Author

You may also provide optional suggestions and comments to authors that they might find helpful in planning their study.

Reviewer #1: The paper was presented as Registered Report Protocol and discusses the research design, methodology and its implementation for estimating the price penalty (defined as willingness to pay) for replacing red meat dishes. In particular, the element that seems to lead the author to investigate the willingness to pay for the replacement of the Western diet is the high environmental impact and unhealthiness of a diet heavily skewed towards red meat consumption. In this regard, in the introduction, the author underlines the lack of evidence in the scientific literature regarding the trade-off that consumers face when asked to change their eating habits.

The study is therefore part of a series of sustainable consumption models, and seeks to highlight a critical aspect that has received far too little attention from the scientific community seeking to propose new models of sustainable production and consumption: the consumer and his habits.

The author assumes that consumers are willing to substitute the meat dishes they normally consume in return for a price penalty (lower willingness to pay) and this is fundamentally acceptable given that the target of the research also aims to highlight the willingness to pay on the part of frequent red meat consumers, therefore with very well rooted eating habits.

My main concern is how the environmental aspect was considered in the study. In fact, the author states that "The research objectives are complemented by exploratory research on the relationship of consumer characteristics with a preference for a substitution option and a calculation of what CO2 taxing level would flip the preference structure if any". The environmental aspect, which was heavily emphasised in the introduction of the work, takes a back seat and becomes a complementary element. This also emerges in the paragraph on the definition of attributes, where the author states that the choice set includes only three attributes: dish, price and waiting time.

Certainly the dish must be one of the attributes, just as the price is essential in determining willingness to pay. However, the choice of the third attribute, waiting time, even if as the author states "represents the value of time which matters particularly in lunch settings and hints at a convenience factor of interest to all kinds of food interventions" should be evaluated more thoroughly.

In my opinion, given that the whole study is designed to assess the link between sustainable consumption patterns and consumer willingness to adopt them, one of the three attributes (if the author does not consider it appropriate to add a fourth) should clearly emphasise the environmental qualities of the substitute dishes, otherwise the consumer's choice of preferred combination of attributes is completely disconnected from the environmental and health characteristics on the basis of which the need to replace the use of red meat in dishes arises.

This aspect, highlighted by the author in Table 1, should be clearly explained to the consumer. In the paragraph on the choice procedure, it is stated that the dishes are explained neutral, however it should be pointed out which aspect of the dishes was explained: only the main ingredients or also the related CF data? I know this may seem like a quibble but it isn't because it would place the study clearly and unequivocally in the context in which the author stated he wanted to place the study from the very first lines of the introduction.

In this same vein, the author should better explain how he wants to relate the results of the IACBC to those of the Carbon Footprint calculation and thus the carbon tax. How might these relationships affect the preference structure and why?

The paragraphs on Sampling procedure, Choice procedure, IACBC and data analysis are well structured, clear and provide the reader with all the necessary information to understand how the methodology under study will be implemented.

As a suggestion for the improvement of the study I suggest the author to better investigate the scientific literature on the application of LCA methodology to different diets, because the studies cited are a bit 'dated and much has been published in this direction by the scientific community.

Reviewer #2: The paper discusses consumer preferences on the substitution of red meat used in the preparation of different food receipts. Despite the interest of the topic, the paper suffers from severe limitations because it does not represent a complete analysis and is lacking from results.

1 title The authors should rethink about the title; maybe better not to mention the examples (red-meat dishes) and just highlight their focus on red meat. However, this is only a personal idea (that does not mean they should do it).

2 Introduction Globally, the introduction should be restructured and rephrased.

3 line 25 the first sentence can be better explained and referenced; alternatively, it should be better to remove it.

4 lines 25-33 better to rephrase, highlighting more who says what (references).

5 lines 25-42 From the beginning, the authors should clearly state and justify their focus on Germany.

6 lines 49-50 "Consumer choices, however, occur on a dish or ingredient level, since the majority of consumers is not committed to a specific diet." Are the authors sure of what they mention? Can they provide a reference for this statement? Honestly, I am not sure this is true (e.g. think about the Mediterranean diet). This cannot be generalised. The authors should better specify it with a reference.

7 line 56 "Additionally, in line with a review by Boer and Aiking (3), this study accounts for the multitude of options to substitute". At the end of this sentence, I would ask: "to do what?". I mean that the authors here introduce this sentence to only partially explain what they will do (within the research). I think the aim should be better and clearly stated.

8 lines 56-60 Where do these options come from? A previous focus group? An existing study? It should be explained to justify them.

9 line 60 Here, the authors mention Spaghetti Bolognese for the first time, without explaining why they mention this. It is not clear for the reader why.

10 lines 60-63 The same of comment n. 8 (please, see above)

11 line 67 Please, remove "I"

12 lines 67-68 This should proceed with what stated in lines 56-63.

13 Theoretical background: I wonder why these four substitutes are being applied to the German sample. Are there any similar studies, before this, dealing with a similar topic on this sample, showing the existing of a knowledge gap that this study could/might fill? I have the idea that the choice of the sample is not justified at all. Moreover, the paragraph globally should be revised, improved both with literature and own description of what the authors are presenting.

14 lines 77-79 Please, improve this part with a more in-depth literature analysis on consumer behaviour on substitution preferences (see your statement: line 54). This is important before starting to describe the four types of substitutes you consider in your study.

15 line 111 Please, get rid of "I". Moreover, what is the role of "whether" in that sentence?

16 line 112 I think the authors mean the four substitutes ("for each dish"). As the authors previously stated, CO2 footprints are based on estimations by IFEU-institute. So, what have the authors done here? Just compared the CO2 footprints of Spaghetti Bolognese, beef topping for 2 buns with butter, and beef meatballs with rice and peas with the relative 4 substitutes? Or have they previously measured the CO2 footprints for the substitutes, before comparing these with that retrieved from IFEU. This is not clear to me.

17 Research objective In this paragraph, the authors state their hypothesis but these are not supported by any relevant previous reference in the literature. It seems to me that such a kind of analysis is a bit self-referential, probably it does not fit the scope of a scientific publication at all.

18 lines 122-123 "Empirical evidence on the WTP for substitution pathways is scarce". I suggest the authors use this statement also within the previous paragraph, to help them explain why they are doing this kind of research.

19 line 124 Please, get rid of "I".

20 Study design: The last part of this paragraph should be stated previously, to justify the research (e.g. describing possible implication, although very briefly).

21 Variables: In this paragraph, the authors should explain all the variables they mention to consider in their research (explanation is provided only for out-of-home consumption habits and consumption habits). Moreover, a table would be used to report all the items and scales. Moreover, the scale mentioned in line 159 should be the opposite (or did the authors reverse the scale? In this case, it should be explained). This paragraph does not provide enough information and should be deeply improved.

22 line 148 Please, get rid of "I".

23 lines 185-200 The authors should clearly state why they chose those three attributes (e.g. inspired by any relevant paper?). Moreover, both the attributes and their levels should be included in a table: this could make it easier for the reader to understand.

24 Results and conclusions are missing in the paper.

7. PLOS authors have the option to publish the peer review history of their article (what does this mean?). If published, this will include your full peer review and any attached files.

Reviewer #1: No

Reviewer #2: No

---

## [Author Response · Author response to Decision Letter 0]

22 Mar 2021

Please refer to the formatted response at the end of the PDF-file

---

## [Decision Letter · Decision Letter 1]

20 May 2021

The price penalty for red meat substitutes in popular dishes and the diversity in substitution

PONE-D-20-36544R1

Dear Dr. Lemken,

We’re pleased to inform you that your manuscript has been judged scientifically suitable for publication and will be formally accepted for publication once it meets all outstanding technical requirements.

Kind regards,

Damian Adams

Academic Editor

PLOS ONE

Additional Editor Comments (optional):

Thank you very much for submitting your revised protocol. Based on reviewer comments, I am happy to accept the protocol at this stage.

Reviewers' comments:

Reviewer's Responses to Questions

**Comments to the Author**

1. Does the manuscript provide a valid rationale for the proposed study, with clearly identified and justified research questions?

Reviewer #1: Yes

Reviewer #2: Yes

2. Is the protocol technically sound and planned in a manner that will lead to a meaningful outcome and allow testing the stated hypotheses?

Reviewer #1: Yes

Reviewer #2: Yes

3. Is the methodology feasible and described in sufficient detail to allow the work to be replicable?

Reviewer #1: Yes

Reviewer #2: Yes

4. Have the authors described where all data underlying the findings will be made available when the study is complete?

Reviewer #1: Yes

Reviewer #2: Yes

5. Is the manuscript presented in an intelligible fashion and written in standard English?

Reviewer #1: Yes

Reviewer #2: Yes

6. Review Comments to the Author

You may also provide optional suggestions and comments to authors that they might find helpful in planning their study.

Reviewer #1: The authors carried out a thorough and valuable revision of the paper. All the critical issues that emerged from the analysis of the first version of the paper have been resolved and the paper is now ready for publication in PLOS ONE. I suggest the authors to make a careful analysis of all the text, tables and figures to correct any typos that become difficult to detect in track change mode.

Reviewer #2: The author/s improved the paper answering point to point to all the comments.

I am satisfied with current version of the paper.

7. PLOS authors have the option to publish the peer review history of their article (what does this mean?). If published, this will include your full peer review and any attached files.

Reviewer #1: No

Reviewer #2: No

---

## [Editor Report · Acceptance letter]

25 May 2021

PONE-D-20-36544R1 

The price penalty for red meat substitutes in popular dishes and the diversity in substitution 

Dear Dr. Lemken:

I'm pleased to inform you that your manuscript has been deemed suitable for publication in PLOS ONE. Congratulations! Your manuscript is now with our production department. 

Kind regards, 

on behalf of

Dr. Damian Adams 

Academic Editor

PLOS ONE